# REVERSIBLE DECOUPLING NETWORK FOR SINGLE IMAGE REFLECTION REMOVAL

## ABSTRACT

Recent deep-learning-based approaches to single-image reflection removal have shown promising advances, primarily for two reasons: 1) the utilization of recognition-pretrained features as inputs, and 2) the design of dual-stream interaction networks. However, according to the Information Bottleneck principle, high-level semantic clues tend to be compressed or discarded during layer-by-layer propagation. Additionally, interactions in dual-stream networks follow a fixed pattern across different layers, limiting overall performance. To address these limitations, we propose a novel architecture called Reversible Decoupling Network (RDNet), which employs a reversible encoder to secure valuable information while flexibly decoupling transmission- and reflection-relevant features during the forward pass. Furthermore, we customize a transmission-rate-aware prompt generator to dynamically calibrate features, further boosting performance. Extensive experiments demonstrate the superiority of RDNet over existing SOTA methods on five widely-adopted benchmark datasets. Our code will be made publicly available.

## 1 INTRODUCTION

Reflection is a common superimposition factor when photographing through transparent medium, such as glass. Under the circumstances, the captured image $I$ typically contains a mixture of transmission $T$ (the scene behind medium) and reflection $R$ (the reflected scene) (Nayar et al., 1997), which can be simply expressed as $I = T + R$. The presence of reflections often hinders vital information in the transmission layer, impeding the performance of downstream computer vision tasks, such as stereo matching, optical flow, and depth estimation (Tsin et al., 2003; Yang et al., 2016; Costanzino et al., 2023). Thus, single image reflection removal/separation (SIRR) is desired to disentangle the transmission and reflection components from a single input image. However, this problem is severely ill-posed as infinitely many possible decompositions of $\hat{T}$ and $\hat{R}$ satisfy $I = \hat{T} + \hat{R}$. In other words, it is highly challenging to determine which combination is optimal if without effective priors or guidance on decomposition.

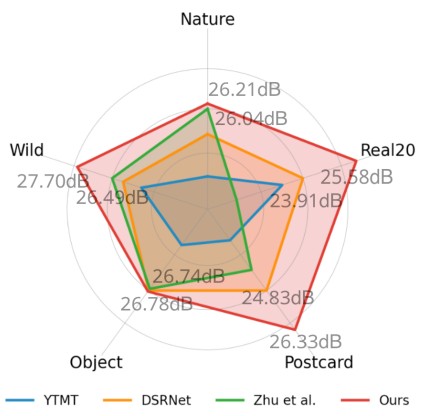

Figure 1: Quantitative comparison in PSNR between ours and previous SOTA methods, where we achieve new records on all 5 datasets. Note that the scale of each axis is normalized by its second-best value. The best and second-best PSNR are displayed for reference.

In recent years, learning-based approaches have made tremendous strides in this field (Zhang et al., 2018; Wei et al., 2019; Li et al., 2023; Hu & Guo, 2023; Zhong et al., 2024). A key consensus among these methods is to exploit hierarchical semantic representations through large-scale recognition-pretrained models, which serve as priors or regularizers during the decomposition. One pioneering deep-learning work (Zhang et al., 2018) leverages intermediate features from a pre-trained VGGNet (Simonyan & Zisserman, 2015) through the concept of hypercolumns to help differentiate between the transmission and reflection layers from mixtures. Originally from neuroscience, the term "hypercolumn" refers to a functional unit in the visual cortex that processes visual stimuli at multiple receptive-field sizes (Hubel & Wiesel, 1974). This concept was first applied to object segmentation and localization by interpolating and stacking features extracted from

different layers of a network (Hariharan et al., 2015b). However, simply mapping stacked high-dimensional hierarchies into a group of much lower-dimensional features–as input for subsequent processes–inevitably leads to considerable semantic information loss.

Previous works with SOTA performance (Hu & Guo, 2021; 2023) suggest that, all information from the source image is valuable for the task. The two components can be optimized by exchanging information between them. For any feasible decomposition $(\hat{T}, \hat{R})$, the following relationship holds:

$$\hat{T} := T - Q, \quad \hat{R} := R + Q \quad \text{s.t.} \quad I = \hat{T} + \hat{R}, \tag{1}$$

where $Q$ represents the information to exchange. Concretely, YTMT (Hu & Guo, 2021) and DSR-Net (Hu & Guo, 2023) select $Q$ via activation functions and channel splitting, respectively. Though being effective, the information preservation is not fully guaranteed in their interaction designs, *i.e.*, the information bottleneck induced by linear layers in YTMT and the multiplicative reductions in the gating mechanism of DSRNet.

To avoid the above risk, reversible units (Gomez et al., 2017b), which are designed to preserve information, may offer a viable solution. In particular, building coupled reversible units naturally fits the situation as follows:

$$\underline{forward\ process} \begin{cases} \hat{T}_2 := \hat{T}_1 + \mathcal{F}(\hat{R}_1) \\ \hat{R}_2 := \hat{R}_1 + \mathcal{G}(\hat{T}_2) \end{cases}; \quad \underline{reverse\ process} \begin{cases} \hat{T}_1 := \hat{T}_2 - \mathcal{F}(\hat{R}_1) \\ \hat{R}_1 := \hat{R}_2 - \mathcal{G}(\hat{T}_2) \end{cases}, \tag{2}$$

where $\mathcal{F}(\cdot)$ and $\mathcal{G}(\cdot)$ can be any network modules, and the subscripts stand for the different versions of layer estimations before and after the reversible units, respectively. For simplicity, we also use $\hat{T}$ and $\hat{R}$ to represent the corresponding deep features, which is based on the understanding that if the features are sufficiently disentangled, mapping them back to the image space becomes an easy task.

Although the use of reversible modules can address the issue of information loss in feature interactions at the same scale, preserving multi-scale information during the feedforward process remains a challenge. Beyond the hypercolumn (Zhang et al., 2018) and the progressive hierarchy fusion (Hu & Guo, 2023), one intuitive scheme is to stack reversible modules at each scale to facilitate forward propagation while incorporating cross-scale connections to ensure effective multi-scale interaction and fusion. A straightforward approach aligning with this idea is MAXIM (Tu et al., 2022) (without consideration of information loss), which employs a fully connected mechanism across multi-scale hierarchies. Similar ideas can also be found in HRNet (Sun et al., 2019). However, operating on high-dimensional features is computationally expensive and memory-intensive.

Inspired by GLOM (Hinton, 2023), which employs a part-whole hierarchy to represent an image with multiple columns, and embodies both bottom-up and top-down interactions to mitigate the computational burden associated with fully connected layers,we integrate multi-scale feature processors into a single sub-network, referred to as a "column". Further, we ensemble the columns in parallel and build interactions in both bottom-up and top-down manners. It is worth noting that, the scaled residual connections used in GLOM for same level interactions between adjacent columns can still cause information loss. To remedy this problem, we extend the residual connections by incorporating multi-level reversible connections, which upgrades the vanilla reversible unit (Gomez et al., 2017a).

Compared with structural designs guided by information bottleneck principle (Tishby & Zaslavsky, 2015; Hu & Guo, 2021), our proposed framework learns disentangled representations (Desjardins et al., 2012; Bengio et al., 2013) by categorizing and recombining the original information, instead of merely selecting and discarding elements, based upon a solid foundation for its information-preserving module (reversible unit). Additionally, it retains multi-scale information and facilitates cross-scale interaction. Besides, in real-world scenarios, the reflection pattern varies along with multiple factors, such as the refractive index of the transparent surface, color granularity, and viewing angle (Schechner et al., 1999). To enhance the robustness against variations in reflection strength, we further endow the model with an adaptive transmission-rate-aware prompt generator.

In light of these considerations, this paper proposes a network, called **R**eversible **D**ecoupling **Net**work (RDNet for short). The major technical contributions of this work are twofold:

- We revisit the preservation and cross-level interaction problems of hierarchical semantic information during the single image reflection removal/separation. To address the challenges, we introduce a multi-column reversible encoder based on the part-whole hierarchy,

complemented by a tailored hierarchy decoder. This design ensures a better retention of rich semantics, effectively mitigating the ill-posed nature of the SIRR task.

- To tackle the varied reflection parameters in real-world scenarios, we introduce an adaptive transmission-rate-aware prompt generator, which learns channel scaling factors from the dataset during training and leverages this knowledge as a prior when testing. It guides the decomposition network in selecting more accurate transmission-reflection ratios, significantly enhancing the model's generalization capabilities.

Extensive experiments are conducted to verify the efficacy of our design, and reveal its superiority over other SOTA alternatives both qualitatively and quantitatively (see Fig. 1 for a brief summary). Notably, our approach also achieves robust generalization on in-the-wild cases, underscoring its practical value in real-world applications (shown by Fig. 4).

## 2 RELATED WORK

### 2.1 SINGLE IMAGE REFLECTION REMOVAL

**Physical formulation.** In prevalent reflection removal frameworks (Levin & Weiss, 2007), an image $I$ is typically decomposed into transmission $T$ and reflection $R$ components, so as to $I = T + R$. However, in real-world scenarios, these two layers may be attenuated by factors such as diffusion and other environmental influences during superposition (Wan et al., 2020). To account for such complexities, an augmented modeling has been proposed: $I = \alpha T + \beta R$, where the coefficients $\alpha$ and $\beta$ provide adaptability to varying conditions (Wan et al., 2018b; Yang et al., 2018). Nonetheless, the assumption of linear superimposition often breaks down, particularly in cases of overexposure (Wen et al., 2019). To address this concern, the concept of an alpha-matting map $W$ is incorporated, leading to a reformulation of the model as $I = W \circ T + \overline{W} \circ R$ with $\overline{W} = 1 - W$. While the adjustment improves the model's flexibility, it also increases the complexity of the already ill-posed problem.

The above model struggles to encapsulate the diverse reflection phenomena, highlighting the challenge of developing a universal solution. Hu and Guo (Hu & Guo, 2023) offered a more comprehensive depiction of the superimposition process by introducing a residual term: $I = \tilde{T} + \tilde{R} + \phi(T, R)$, where $\tilde{T}$ and $\tilde{R}$ signify the altered transmission and reflection information within $I$ after superimposition and degradation, as captured by camera sensors. The term $\phi(T, R)$ denotes the residual information in the reconstruction, arising from factors such as attenuation and overexposure. However, current methods primarily use the above modelings to synthesize training data, expecting the generalizability to real-world data. But, they lack explicit estimation of the physical parameters involved. Furthermore, distance-based loss functions such as mean absolute error (MAE) and mean squared error (MSE) fail to account for global color and intensity shifts. Explicitly estimating the degradation rate of the projected image could improve performance. A more detailed explanation is provided in Sec. 3.2.

**Deep-learning-based modeling.** Considering that reflection layers are typically out of focus and appear more blurred than transmission layers, Li and Brown (Li & Brown, 2014) introduced a relative smoothness prior to distinguish the gradients of the two layers, which follow different probability distributions. Multi-scale depth-of-field (DoF) analysis-based methods were also developed to separate reflections from transmissions by detecting reflection-dominated regions (Wan et al., 2018a). While these approaches achieved promising results in well-controlled environments, their performance significantly drops in real-world conditions. CEILNet (Fan et al., 2017) imposes a relative smoothness prior on synthesizing reflection layers, and combines them with transmission layers through addition. It introduces an edge-aware network designed to capture transmission components, but it neglects high-level semantics, which could further enhance the SIRR task. These methods with hand-crafted priors highly likely fail in challenging real-world cases.

Zhang *et al.* (Zhang et al., 2018) enhanced semantic awareness by leveraging hypercolumn features extracted from a pre-trained VGG-19 network (Hariharan et al., 2015a), together with perceptual and adversarial losses. ERRNet (Wei et al., 2019) uses misaligned pairs as training data to take a step further. But it overlooks the reflection layer, potentially increasing ambiguity in transmission recovery. Li *et al.* (Li et al., 2023) proposed RAGNet, a two-stage network that initially estimates the reflection component and then uses it to guide transmission prediction. Recently, the YTMT strategy proposed in (Hu & Guo, 2021) treats both components equally through a dual-stream interactive network that

restores both layers simultaneously. Yet, noticing the problem hidden in the physical formulation, their interaction module relies on a linear assumption, which may upper-bound its performance. Other methods, such as BDN (Yang et al., 2018) and IBCLN (Li et al., 2020) employ reflection models with scalar weights to iteratively estimate both components, ensuring that the reflection is not too faint. However, the interaction between the two components is ignored, sometimes leading to heavy ghosting effect in transmission and reflection. Dong *et al.* (Dong et al., 2021) developed an iterative network that estimates a probabilistic reflection confidence map at each step. DSRNet (Hu & Guo, 2023) introduces a mutually gated interaction mechanism within a two-stage structural design. In the first stage, the network progressively fuses extracted hierarchical features, while the second stage focuses on further decomposing these features. However, the issue of information loss persists due to the multiplicative reductions in the gating mechanism. Additionally, the progressive hierarchical fusion, isolated in the first stage, does not fully ensure that the hierarchical information is preserved during the subsequent decomposition processes. Zhu *et al.* (Zhu et al., 2024) proposed a maximum reflection filter for estimating reflection locations and introduce a large dataset, but they similarly overlook interaction between the two layers. Our proposed RDNet addresses the drawbacks of existing approaches by incorporating reversible connections and a multi-column design.

## 2.2 Reversible Network

Reversible neural networks are designed to prevent information loss by enabling the recovery of original inputs from outputs, thereby maintaining data integrity. Deco and Brauer (Deco & Brauer, 1994) introduced a reversible architecture that guarantees data preservation through a residual design, which generates a lower triangular Jacobian matrix with unity diagonal elements. Building upon this concept, Dinh *et al.* (Dinh et al., 2015) developed the NICE framework, employing a non-linear bijective transformation between the data and a latent space. However, this design only allows volume-preserving mappings. Dinh *et al.* (Dinh et al., 2017) extended extended this idea by proposing a reversible transformation that does not require volume preservation. While Gomez *et al.* (Gomez et al., 2017a) combined the concept of invertible networks with the ResNet architecture, ensuring that each layer's activations can be derived from the subsequent layer's activations. This manner enables backpropagation without storing the activations in memory, except for a few non-reversible layers.

**Reversible Networks for Low-level Vision.** Reversible CNNs have been effectively applied to various low-level tasks, including compression (Liu et al., 2021), enhancement (Zhu et al., 2022; Wang et al., 2022; Li et al., 2022) and restoration (Huang & Dragotti, 2022; Zhu et al., 2023; Yao et al., 2023). These solutions typically employ reversible networks as a shared encoder-decoder in a generative manner, where new textures are generated to supplement ost information during degradation. However, in the task of reflection removal, the target result (the transmission image) is mixed with the reflection rather than lost. This task requires precise decoupling of the input image components instead of generating new textures. To the best of our knowledge, our work is the first to design a reversible architecture specifically for reflection removal.

## 3 Methodology

In this section, we present the key components of the proposed RDNet, the overall structure of which is schematically depicted in Fig. 2. Specifically, it is composed of three primary modules: the multi-column reversible encoder (MCRE), transmission-rate-aware prompt generator (TAPG) and the hierarchy decoder (HDec). The Pretrained Hierarchy Extractor (PHE) captures semantically rich hierarchical representations from the input image and transmits them to each level of the first column in MCRE. Meanwhile, TAPG learns channel-level transmission-reflection ratio priors from the data, mapping these learned fundamental parameters into prompts that guide the MCRE network. Finally, each column in MCRE employs an HDec to encode the hierarchical information, providing effective side guidance (Qin et al., 2020). The decoded hierarchies from the last column yield the final results.

### 3.1 Multi-scale Reversible Column Encoder

As shown in Fig. 2, our proposed Multi-Column Reversible Encoder (MCRE) employs an architecture that differs from end-to-end models (Zhang et al., 2018; Wei et al., 2019) by incorporating multiple sub-networks, each receiving column embeddings modulated by the Transmission-rate-Aware Prompt

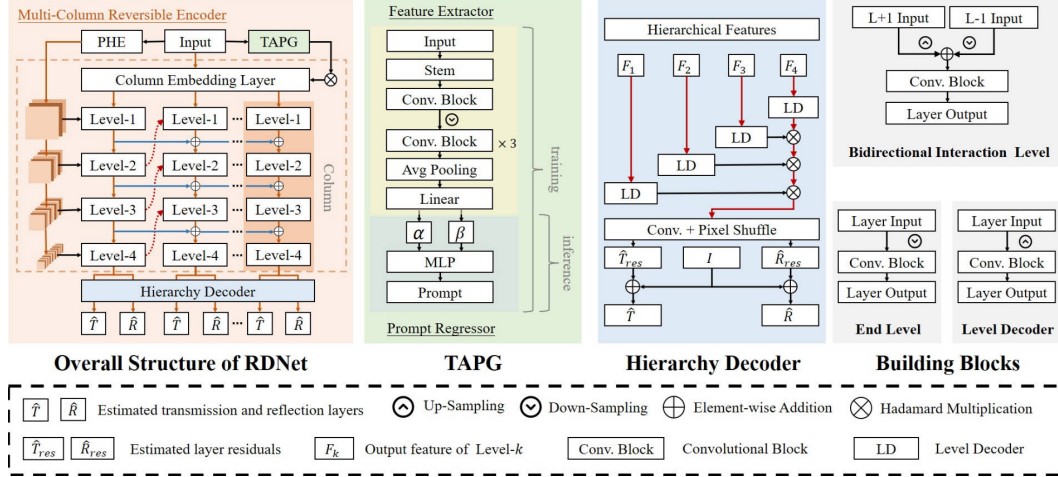

Figure 2: Overall structure of our RDNet, the input is fed in the transmission-rate-aware prompt generator, pretrained hierarchy extractor, and the column embedding. The output of the prompt generator will be transferred into the column network. After interactions between the columns, each column uses a separate decoder to obtain a pair of image layers.

Generator (TAPG). The model is composed of a Column Embedding Layer and multiple columns that encode multi-scale information.

In MCRE, information propagation between columns is handled through two primary mechanisms: intra-level reversible connections (denoted by blue solid lines in the figure) that facilitate information preservation between columns at the same level, and inter-level connections (illustrated as red dashed lines) paired with Bidirectional Interaction Levels, enabling interactions across adjacent levels. This approach effectively decouples multi-scale features up to Level-3. As an exception, Level-4 lacks corresponding cross-level connections, conforming to the structure of the End Level. The initial column within MCRE accepts the hierarchical information extracted by the PHE, ensuring a semantic-rich representation. The subsequent multi-column reversible design ensures the lossless propagation of hierarchical information throughout the decomposition network.

Specifically, our column embedding layer employs a $7 \times 7$ convolution layer with a stride of 2, producing $2 \times 2$ overlapping patches $F_{-1}$ for subsequent processes. For the $i$-th column ($i \in \{1, 2, ..., N\}$), each level feature $F_j^i, j \in \{0, 1, 2\}$ receives information $F_{j-1}^i$ from the lower level of the current column and $F_{j+1}^{i-1}$ from a higher level of the previous one. The collected features are further fused with the signal $F_j^{i-1}$ of the current level. The operation described above for the level $j$ is expressed as:

$$F_j^i = \omega(\theta(F_{j-1}^i) + \delta(F_{j+1}^{i-1})) + \gamma F_j^{i-1}. \tag{3}$$

where $\omega$ denotes the network operation, while $\theta$ and $\delta$ represent downsampling and upsampling operations, respectively. The $\gamma$ term is a simple reversible operation. In our implementation, we utilize a learnable reversible channel-wise scaling as the reversible operation $\gamma$. **This connection is information lossless, as one can retrieve $F_j^{i-1}$ through the reverse operation:**

$$F_j^{i-1} = \gamma^{-1} \left[ F_j^i - \omega(\theta(F_{j-1}^i) + \delta(F_{j+1}^{i-1})) \right]. \tag{4}$$

Notably, for the first level of each column, we define $F_{-1}^i := F_{-1}$. Moreover, since the last level does not receive any higher-level features, the $\delta(F_{j+1}^{i-1})$ term is hence discarded.

**Hierarchy Decoder.** Our hierarchy decoder integrates hierarchical codes from all scales to generate the final output. We leverage several Level Decoders (LD) to interpret the higher-dimensional hierarchies with smaller resolution into lower-dimensional ones at larger resolution. The up-sampling operator in an LD is implemented by pixel-shuffle (Shi et al., 2016), an information-consistent operator before and after the scaling. The up-sampled features are then fused with the information from the previous scale with multiplication modulation. Ultimately, the final LD produces the layer

residuals ($\hat{T}_{res}$ and $\hat{R}_{res}$) through another pixel-shuffle up-sampling operation and are connected with the original input to obtain the layer decomposition $\hat{T}$ and $\hat{R}$.

## 3.2 TRANSMISSION-RATE-AWARE PROMPT GENERATOR

Previous methods for SIRR often exhibit limited generalization capabilities due to the inherent complexity and variability of optical factors in real-world reflective scenarios, compounded by the constraint of limited training data. This limitation can be observed in the real-world test samples we collected, as shown in Fig. 4. Meanwhile, in both real-world and synthesized data, color/intensity is often compromised due to the reflection overlaying the transmission, with the transmission $T$ itself being degraded by a transmission rate $a$. In image restoration tasks, the ground truths are typically clean images. But, linearly deviated input/result often occurs because of color/illumination shifts in the real-world scenarios. The phenomenon is further detailed in the appendix A.

To solve the aforementioned problems, we develop a transmission-rate estimator using a simplified version of the ConvNext model (Liu et al., 2022) pre-trained on ImageNet-1k (Deng et al., 2009). Given an input image $I \in \mathbb{R}^{3 \times H \times W}$, our transmission-rate estimator predicts six parameters: $\alpha_{\{R,G,B\}}, \beta_{\{R,G,B\}}$ such that $\|\alpha_i T + \beta_i - I\|_2$ is minimized for each $i \in \{R, G, B\}$. When testing the input image using the six parameters generated by the prompt generator, we can obtain an average PSNR of **24.34dB** across four benchmark datasets (Real20, Objects, Postcard, and Wild), surpassing the previous state-of-the-art method by Dong *et al.* (Dong et al., 2021). This result confirms the effectiveness of our estimated transmission rate.

Once the transmission rate factor $\alpha_{\{R,G,B\}}, \beta_{\{R,G,B\}}$ is estimated, a three-layer MLP is used to generate prompts that guide the MRCE, resulting in a prompt $P \in \mathbb{R}^{C \times H \times W}$, where $C$ represents the output dimension of the patch embedding layer, set to 64 in our work. Subsequently, the prompt is used to modulate the intermediate features from the column embedding layer $F$ into $P \circ F$, which allows the network to dynamically adapt to the specific characteristics of each input image, thereby enhancing the accuracy of reflection removal.

## 3.3 TRAINING OBJECTIVE

Our model undergoes two training stages. In the first stage, we train the estimator for the transmission rate. Once this is complete, we fix the classifier and proceed to train the main model along with the prompt generator. This training scheme ensures that both the transmission-rate-aware prompt generator and the main model work harmoniously towards the task, resulting in a robust solution.

We employ content loss and perceptual loss for the task, evaluating each pair of images produced by each column using the following loss functions before aggregating them into the final outcome.

**Content Loss.** The content loss ensures consistency between the output images and the ground truth training data. In the image domain, we adopt the Mean Squared Error (MSE) loss. Following previous works (Hu & Guo, 2023; 2021), we further regularize the model by encouraging consistency between the output and ground truth in the gradient domain, which writes:

$$\mathcal{L}_{\text{cont}} := c_0 \|\hat{T} - T\|_2^2 + c_1 \|\hat{R} - R\|_2^2 + c_2 \|\nabla \hat{T} - \nabla T\|_1, \tag{5}$$

where $\|\cdot\|_1$ and $\|\cdot\|_2$ stand for the $\ell_1$ and $\ell_2$ norms, respectively. During the first stage of training, we set $c_0 = 1, c_1 = 0, c2 = 0$. In the second stage, these values are adjusted to $c_0 = 0.3, c_1 = 0.9, c_3 = 0.6$.

**Perceptual Loss.** To enhance the perceptual quality of images produced by our model, we minimize, we minimize the $\ell_1$ discrepancy between the features of predicted elements and the ground-truth references. This comparison is made at the 'conv2_2', 'conv3_2', 'conv4_2', and 'conv5_2' layers of a pre-trained VGG-19 network on the ImageNet dataset. Denoting the features at the $i$th layer as $\phi_i(\cdot)$, the perceptual loss is computed as:

$$\mathcal{L}_{\text{per}} := \sum_j \omega_j \|\phi_j(\hat{T}) - \phi_j(T)\|_1, \tag{6}$$

where $\omega_j$ are weighting coefficients for each layer. The total loss turns out to be:

$$\mathcal{L} := \mathcal{L}_{\text{cont}} + w \mathcal{L}_{\text{per}}, \tag{7}$$

where $w = 0.01$ is empirically set.

Table 1: Quantitative results of various methods on four real-world benchmark datasets. The best results are highlighted in **bold**, and the second-best results are underlined.

| | Methods | Real20 (20) | | Objects (200) | | Postcard (199) | | Wild (55) | | **Average** | |
|---|---|---|---|---|---|---|---|---|---|---|---|
| | | PSNR | SSIM | PSNR | SSIM | PSNR | SSIM | PSNR | SSIM | PSNR | SSIM |
| w/o Nat. | ERRNet | 22.89 | 0.803 | 24.87 | 0.896 | 22.04 | 0.876 | 24.25 | 0.853 | 23.53 | 0.879 |
| | IBCLN | 21.86 | 0.762 | 24.87 | 0.893 | 23.39 | 0.875 | 24.71 | 0.886 | 24.10 | 0.879 |
| | RAGNet | 22.95 | 0.793 | 26.15 | 0.903 | 23.67 | 0.879 | 25.53 | 0.880 | 24.90 | 0.886 |
| | YTMT | 23.26 | 0.806 | 24.87 | 0.896 | 22.91 | 0.884 | 25.48 | 0.890 | 24.05 | 0.886 |
| | DSRNet | 24.23 | 0.820 | **26.28** | **0.914** | 24.56 | 0.908 | 25.68 | 0.896 | 25.40 | 0.905 |
| | Ours | **24.43** | **0.835** | 25.76 | 0.905 | **25.95** | **0.920** | **27.20** | **0.910** | **25.95** | **0.908** |
| w Nat. | Dong *et al.* | 23.34 | 0.812 | 24.36 | 0.898 | 23.72 | 0.903 | 25.73 | 0.902 | 24.21 | 0.897 |
| | DSRNet | 23.91 | 0.818 | 26.74 | 0.920 | 24.83 | 0.911 | 26.11 | 0.906 | 25.75 | 0.910 |
| | Zhu *et al.* | 21.83 | 0.801 | 26.67 | **0.931** | 24.04 | 0.903 | 26.49 | 0.915 | 25.34 | 0.912 |
| | Ours | **25.58** | **0.846** | **26.78** | 0.921 | **26.33** | **0.922** | **27.70** | **0.915** | **26.65** | **0.917** |

Table 2: Quantitative results on the Nature dataset. The competitors are all trained with the additional data from the Nature dataset

| | ERRNet | IBCLN | YTMT | DSRNet | Zhu *et al.* | Ours |
|---|---|---|---|---|---|---|
| PSNR | 22.18 | 23.57 | 23.85 | 25.22 | 26.04 | **26.21** |
| SSIM | 0.756 | 0.783 | 0.810 | 0.832 | **0.846** | 0.842 |

## 4 EXPERIMENTAL VALIDATION

### 4.1 IMPLEMENTATION DETAILS

Our model is implemented in PyTorch (Paszke et al., 2019) and optimized with Adam optimizer (Kingma & Ba, 2015) on an RTX 3090 GPU for 20 epochs. The learning rate is initialized at $10^{-4}$, and remains fixed throughout the training phase, with a batch size of 2. The training dataset comprises both real and synthetic images. To align with previous works, we evaluate the performance of our model under two commonly used data settings: a) The setting from (Hu & Guo, 2021; Wei et al., 2019) and (Li et al., 2023), which consists of 90 real image pairs from (Zhang et al., 2018) and 7,643 synthesized pairs from the PASCAL VOC dataset (Everingham et al., 2010); and b) The setting from (Hu & Guo, 2023) and (Dong et al., 2021), which includes 200 additional real image pairs provided by (Li et al., 2020). For data synthesizing, we follow the pipeline and physical model from DSRNet (Hu & Guo, 2023), represented by $I = \alpha T + \beta R - T \circ R$. Slightly, we modify this approach by sampling individual $\alpha$ and $\beta$ for R, G, and B channels. This adjustment aims to prevent the transmission rate estimator from converging to a trivial solution. The parameters the PHE are initialized by a pretrained FocalNet (Yang et al., 2022).

### 4.2 PERFORMANCE EVALUATION

For the comparison, we evaluate seven state-of-the-art methods: ERRNet (Wei et al., 2019), IBCLN (Li et al., 2020), RAGNet (Li et al., 2023), Dong *et al.* (Dong et al., 2021), YTMT (Hu & Guo, 2021), DSRNet (Hu & Guo, 2023), Zhu *et al.* (Zhu et al., 2024), on four real-world datasets, including Real20 (Zhang et al., 2018) and three subsets of the SIR$^2$ Datasets (Wan et al., 2017), for the Nature Dong et al. (2021) dataset, we compare IBCLN, ERRNet, YTMT, DSRNet and Zhu *et al.*.

**Quantitative comparisons.** The quantitative result is shown in Tab. 1. We directly employ the code and pre-trained weights publicly provided by their authors to obtain all the quantitative results. To make a fair comparison, the methods with and without additional data from the Nature dataset are compared separately. Apparently, our methods show their superiority over other competitors on all testing datasets, only falling short on SSIM compared to Zhu *et al.* on the Objects dataset. Our methods achieved a promising boost, especially on the Real20 dataset, which contains hard cases collected in real-world conditions, meaning our method can better fit real-world conditions. The other three datasets contain a variety of scenes, illumination conditions, and glass thickness, meaning our

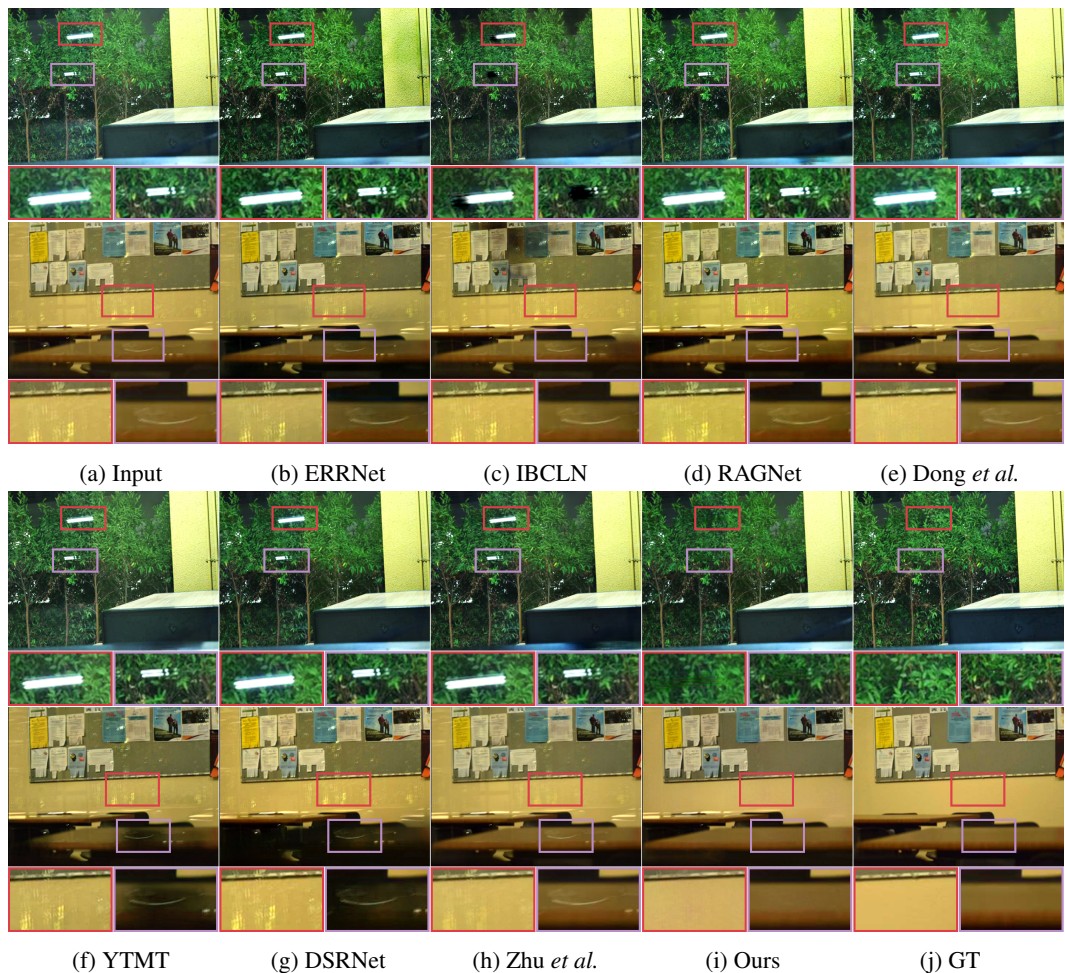

(a) Input     (b) ERRNet     (c) IBCLN     (d) RAGNet     (e) Dong *et al.*

(f) YTMT     (g) DSRNet     (h) Zhu *et al.*     (i) Ours     (j) GT

Figure 3: Qualitative comparisons on samples from the Wild dataset. Please zoom in for more details. More visual results can be found in the appendix.

method performs better in most conditions. The experimental result demonstrates that our proposed SIRS method can adapt to complicated situations and has a stronger generalization ability.

For a comprehensive comparison, we present the results obtained on the Nature dataset in Tab. 2, which comprises 20 real-world samples. Our method achieved the best PSNR and the second-best SSIM, with a marginal decrease of only 0.004 in SSIM. These results further underscore the superiority of our approach in real-world scenarios.

**Qualitative comparisons.** The qualitative comparison is shown in Fig. 3 and Fig. 4, with additional visual examples provided in the appendix. The first case in Fig. 3 illustrates a highly reflective object, which presents a significant challenge for reflection removal techniques due to its intensity. Our method successfully eliminates the reflective object, accurately revealing the underlying texture and color information. This performance is superior to other methods, highlighting our approach's effectiveness in handling complex real-world reflections. In contrast, ERRNet, RAGNet, Dong *et al.*, YTMT, DSRNet and Zhu *et al.* struggle to remove the object, leaving it almost entirely intact. Although IBCLN partially removes the reflection, it fails to recover the underlying color information, resulting in an incomplete outcome. This example clearly demonstrates our method's advanced capability in accurately identifying and removing even strong and complex reflections, further proving its robustness in real-world scenarios.

The second example further showcases our method's proficiency in handling reflections spread across an image. Here, the reflection is complex and covers a large area, which other methods fail to remove

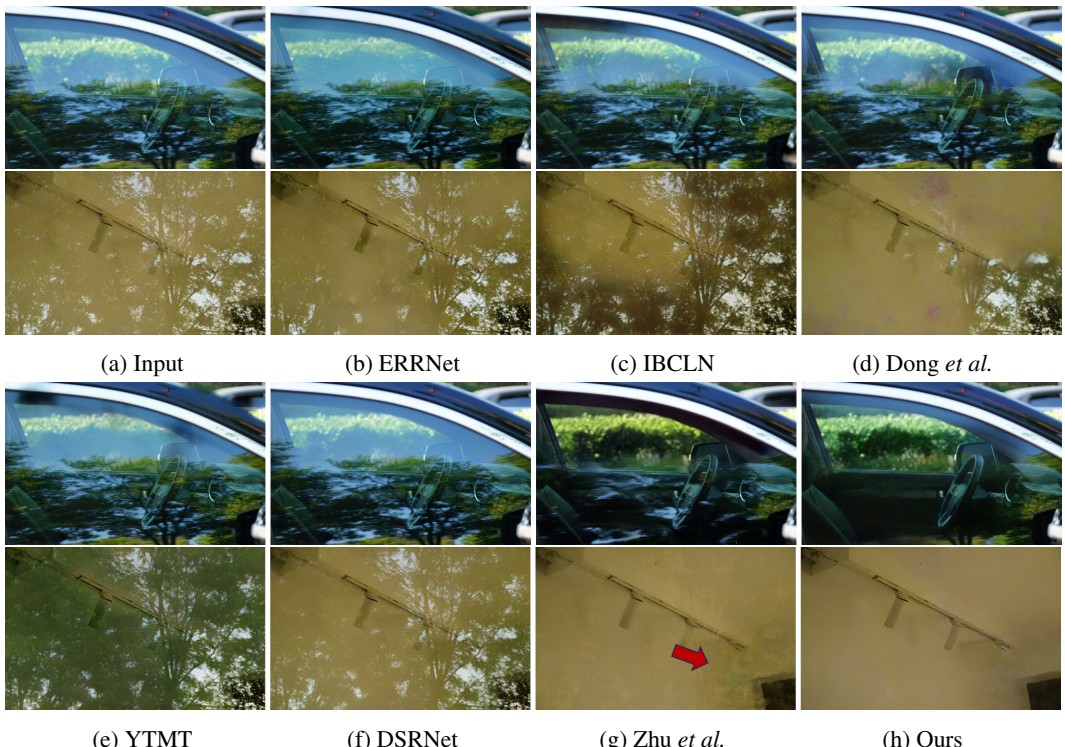

| (a) Input | (b) ERRNet | (c) IBCLN | (d) Dong *et al.* |
| (e) YTMT | (f) DSRNet | (g) Zhu *et al.* | (h) Ours |

Figure 4: Qualitative comparisons on real-world cases. Please zoom in for more details.

effectively. In contrast, our approach accurately targets and eliminates the majority of the reflection, preserving the integrity of the non-reflective elements.

Figure 4 illustrates the robustness of our method in real-world scenarios. These two cases were captured in real-life conditions by us. In the first example, a dense reflection covers the car window, a challenge that competing methods largely fail to address, with only Zhu *et al.* managing partial removal. However, our approach almost entirely separates the reflections, producing more visually appealing results. A similar outcome is observed in the second example, where our method successfully removes nearly all reflections. In contrast, all other methods struggle to handle this scenario effectively. These examples demonstrate the robustness of our decoupling paradigm, confirming its effectiveness in real-world scenarios.

These results demonstrate the effectiveness of our decoupling routine, offering several key advantages: 1) accurate identification and separation of reflection components from underlying content, 2) robust performance in removing dense reflections common in real-world scenarios, and 3) strong generalizability across diverse conditions. Collectively, these findings validate the theoretical soundness and practical efficacy of our proposed method.

### 4.3 ABLATION STUDIES

To better verify the effect of our prompt generator and reversible network structure, we bring a series of ablation studies, including different settings of network structure and prompt generator. The results are gathered in Tab. 3. We present the results of our prompt generator on the left side and the results of our network structure on the right side.

**Discussion on transmission-rate-aware prompt generator.** To inform the model with the transmission rate, a straightforward approach is to adjust the input image to enhance it globally using the estimated transmission rate. Specifically, for $I := aT + bR + \phi(T, R)$, we adjust the input $I$ to $\frac{1}{a}I := T + \frac{b}{a}R + \frac{1}{a}\phi(T, R)$. This operation is denoted as Pre. in Tab. 3. As shown in Tab. 3, if we remove all transmission-rate-aware techniques (setting A), the average performance drops by 1.13

Table 3: Ablation studies on the prompt generator and different network configurations.

| Setting | Prompt | Pre. | Average | | Setting | Dual-stream | Ref. Loss | Invertibility | Average | |
|---------|--------|------|---------|---------|---------|-------------|-----------|---------------|---------|---------|
| | | | PSNR | SSIM | | | | | PSNR | SSIM |
| A | ✗ | ✗ | 25.52 | 0.909 | D | ✓ | ✓ | ✓ | 26.37 | 0.917 |
| B | ✗ | ✓ | 25.99 | 0.910 | E | ✗ | ✗ | ✓ | 25.99 | 0.914 |
| C | ✓ | ✓ | 26.03 | 0.913 | F | ✗ | ✓ | ✗ | 24.05 | 0.884 |
| Ours | ✓ | ✗ | **26.65** | **0.917** | Ours | ✗ | ✓ | ✓ | **26.65** | **0.917** |

dB. If we adopt the straightforward method described above (setting B), the performance recovers by 0.47 dB. This confirms the importance of informing the model with the transmission rate.

However, as we analyzed in Section 3.2, directly adjusting the input image is far from optimal. Due to potential inaccuracies in the estimation in some scenarios, directly adjusting the model can introduce an additional shift that is difficult to correct during second-stage training. A more subtle and flexible approach is to reweight the feature channels with our transmission-rate-aware prompt.

To verify this, we both adjust the input and add a transmission-rate-aware prompt to the feature (setting C). The performance remains nearly the same as in Setting B, indicating that adjusting the input makes it challenging for the model to recover from incorrect estimations. Finally, our model with the proposed transmission-rate-aware prompt outperforms all variants, demonstrating its efficacy.

**Discussion on model design.** To verify the rationality of our design of the decoupling model, we created three new variants of our model. We modify our RDNet to a DSRNet-style one, where two streams estimate transmission and reflection separately in a single column, and interact with each other. This variant is denoted as Dual-stream (Setting D). As shown in Tab. 3, even with double computation, the performance still drops by 0.28 dB. This confirms the superiority of our decoupling design compared to the dual-stream design. Secondly, we removed the reflection part ($c_1 \|\hat{R} - R\|_2^2$) in the content loss function (Eq. (5)), leaving only the transmission part ($c_0 \|\hat{T} - T\|_2^2 + c_2 \|\nabla\hat{T} - \nabla T\|_1$) in the training process. This variant is denoted as Ref. Loss. (Setting E). A performance drop of 0.66dB can be observed. This confirms the necessity of the reflection loss function. Without regularization predicting the other component, the network weakens its ability to clearly identify both components in single-stream feature maps.

To verify the necessity of the invertibility of the network in the reflection removal task, we replace the reversible connection with the U-Net connection (Ronneberger et al., 2015) (Setting F). Although it requires slightly more parameters and much more memory, a massive performance drop of 2.6 dB can be discovered, indicating the importance of invertibility design.

## 5 CONCLUSION

In this paper, we proposed RDNet, a novel model for addressing key challenges in the task of single image reflection removal. Specifically, RDNet tackles the limitations of insufficient utilization of multi-scale, pretrained hierarchical information and information loss during feature decoupling. The multi-column reversible structure enables the preservation of rich semantic features, which are then effectively leveraged in the multi-scale processing of each column. Furthermore, the proposed Transmission-rate-Aware Prompt Generator alleviates the inherent conflict between complex reflection parameters and limited training data. Through these innovations, RDNet demonstrates an enhanced capability for robust reflection removal. Our method demonstrates superior performance compared to state-of-the-art techniques across a range of real-world benchmark datasets, highlighting its robustness and adaptability in diverse reflective scenarios. Ablation studies further validate the effectiveness of our key contributions, confirming the advantages of our design choices. It is positive that our work opens up new avenues for research in reflection removal, and has the potential to impact various applications in computer vision and image processing significantly.

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

# APPENDIX

## A FURTHER DISCUSSION ABOUT THE TAPG

As illustrated in Fig. 5 with a toy visualization of a pure white image, even though the color bias and noise exhibit the same Mean Squared Error (MSE) relative to the ground truth, a linear estimation can instantly correct the image's color shift, whereas noise requires more complex operations to address. Metrics like MSE and Mean Absolute Error (MAE) struggle to compel the network to effectively recognize and rectify the linear degradation in the physical formulations. In this context, by pre-calibrating the features with a transmission-rate-aware prompt, we can significantly mitigate the effects of linear degradation, such as color and intensity inconsistencies.

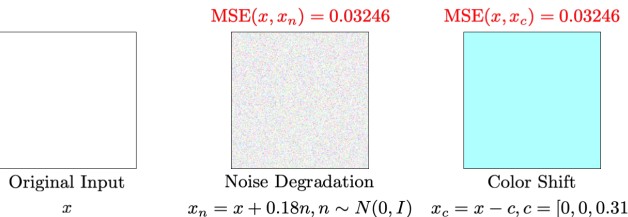

Figure 5: Visualization of the drawback of Mean Squared Error (MSE). Both color shifts and noise degradation exhibit the same MSE relative to the ground truth.

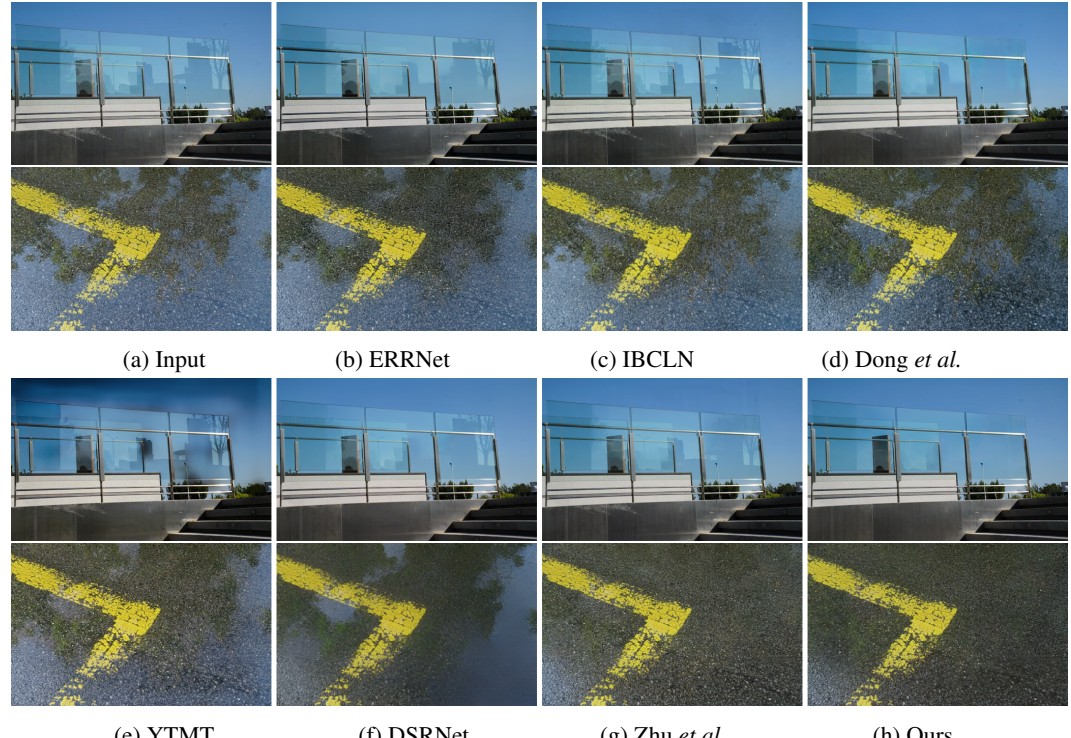

| (a) Input | (b) ERRNet | (c) IBCLN | (d) Dong *et al.* |

| (e) YTMT | (f) DSRNet | (g) Zhu *et al.* | (h) Ours |

Figure 6: Visual comparison of estimated transmission layers between state-of-the-arts and ours on real-world samples.

## B   QUALITATIVE COMPARISONS

**More visual cases.** We exhibit a total of nine additional cases: two cases from the Real20 dataset in Fig. 7, three cases from the Solid dataset in Fig.8, two cases from the Postcard dataset in Fig. 9 and two real-world cases captured by us in Fig. 6. As illustrated, our method excels at revealing the information obscured by reflections and is highly effective in removing the majority of the reflections.

## C   ADDITIONAL EXPERIMENTS

**The ablation study for the number of columns.** In this study, we investigate the effect of varying the number of columns on the overall performance in Tab. 4. Specifically, we adjusted the number of columns after the first PHE column, experimenting with configurations of 2, 4, and 6 columns. Our findings indicate that a configuration with 4 columns yields the highest performance. In contrast, configurations with 2 and 6 columns resulted in performance drops of 0.4dB and 0.46dB in PSNR, respectively. This suggests that an optimal balance exists, where too few or too many columns can detract from the model's performance.

Table 4: The experiment of changing numbers of columns. The best results are indicated in **bold**.

| Num Col | Average | |
|---|---|---|
| | PSNR | SSIM |
| 2 | 26.25 | 0.914 |
| 4 | **26.65** | **0.917** |
| 6 | 26.19 | 0.910 |

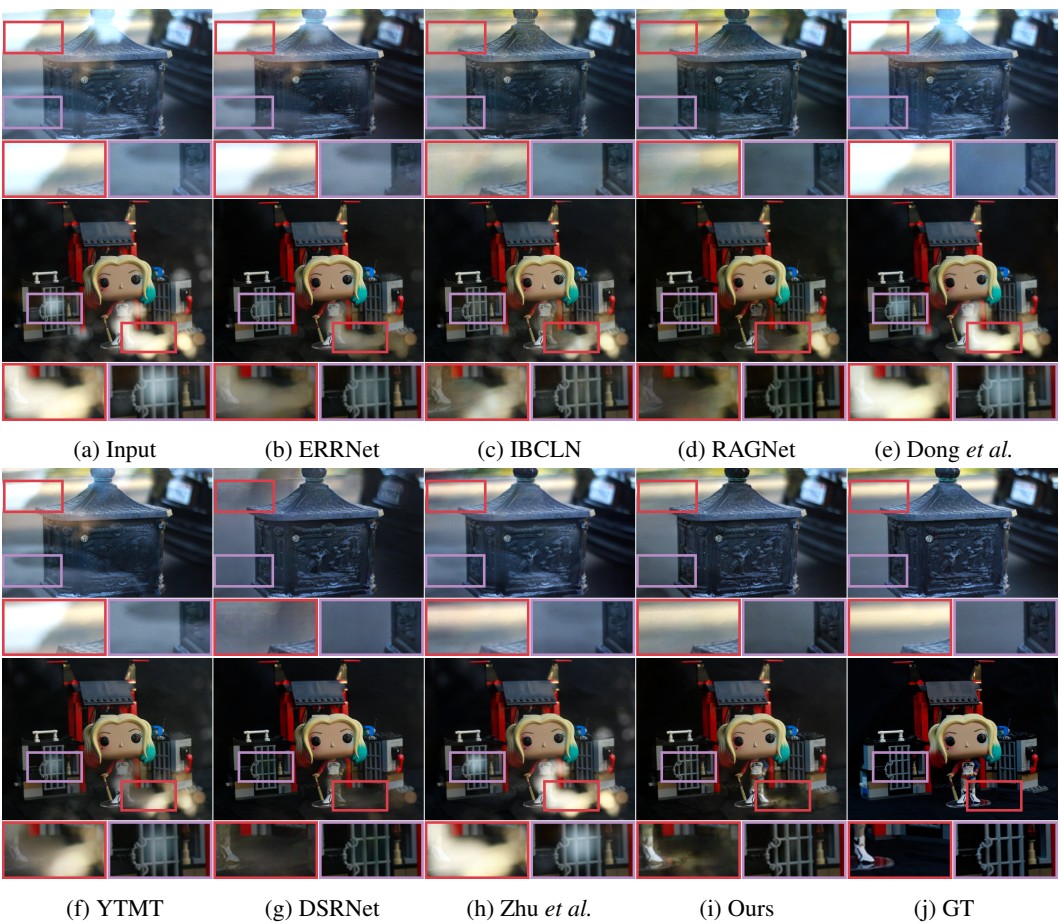

Figure 7: Visual comparison of estimated transmission layers between state-of-the-arts and ours on real-world samples (Real 20).

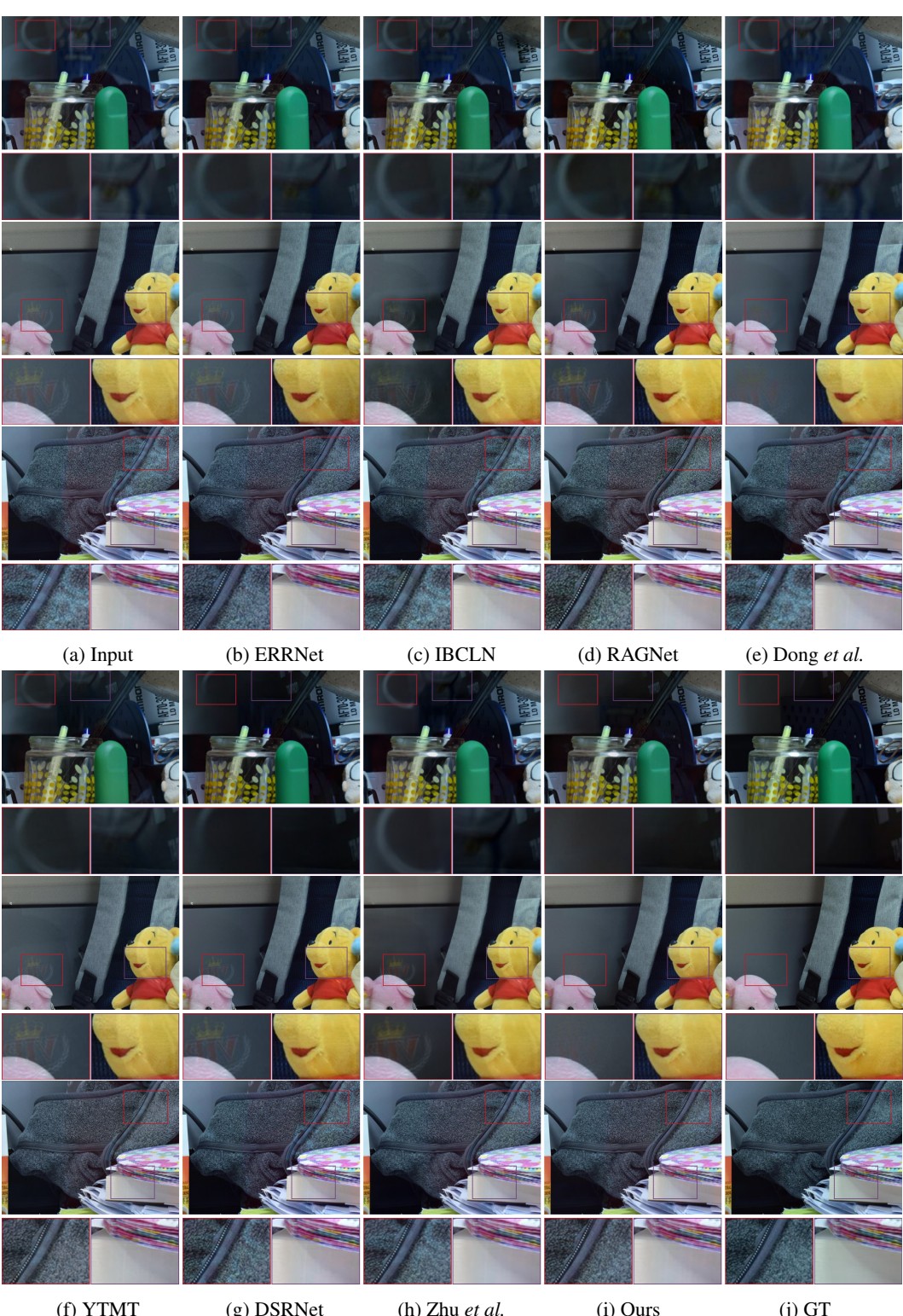

Figure 8: Visual comparison of estimated transmission layers between state-of-the-arts and ours on Objects dataset.

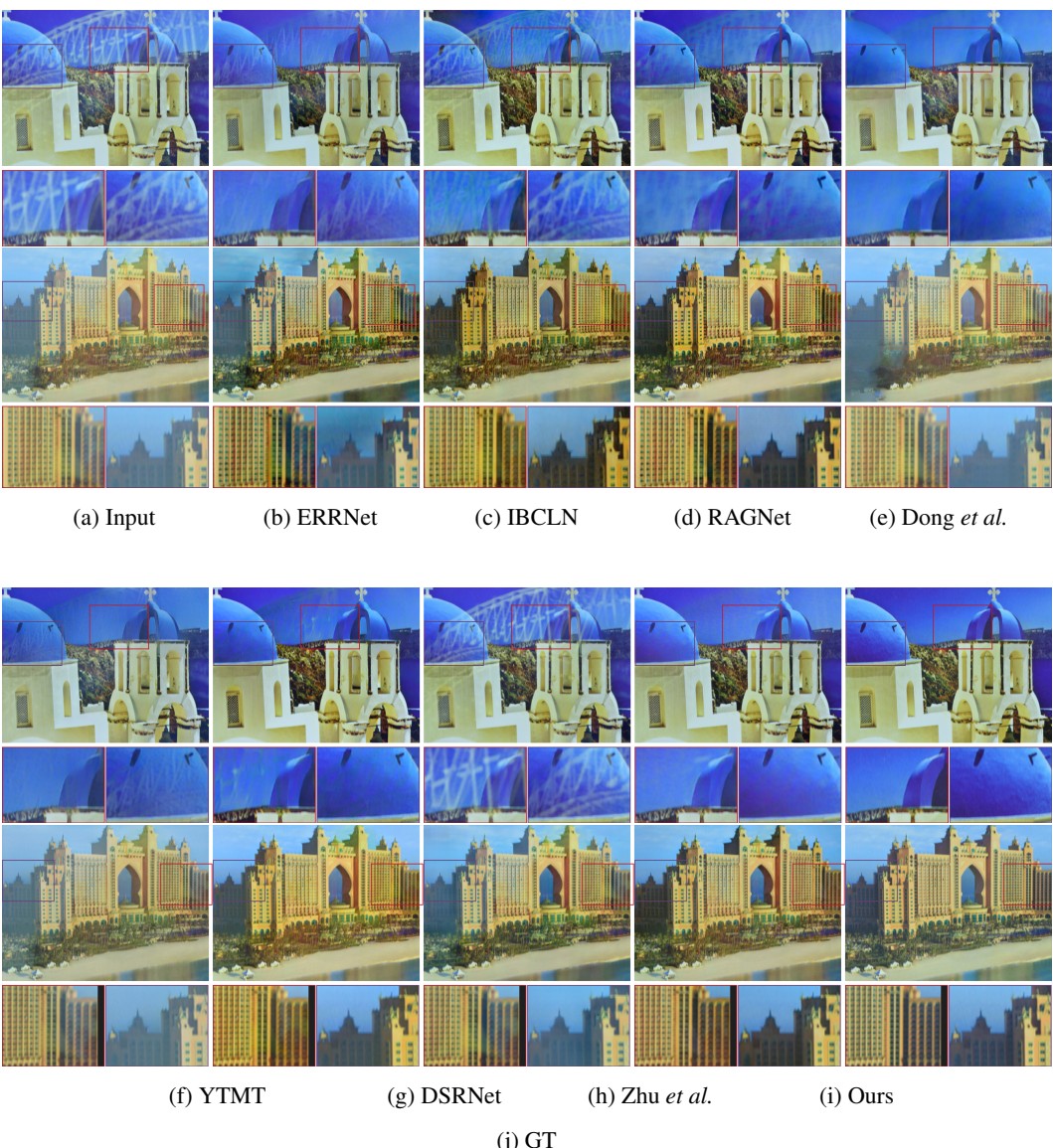

Figure 9: Visual comparison of estimated transmission layers between state-of-the-arts and ours on Postcard dataset.

