# OpenReview forum: "Reversible Decoupling Network for Single Image Reflection Removal"
_ICLR.cc/2025/Conference — ICLR 2025 Conference Withdrawn Submission_

### Official Review · Reviewer_bHU9 · 2024-11-01

**Soundness:** 3
**Presentation:** 2
**Contribution:** 3
**Rating:** 5
**Confidence:** 4

**Summary:**

This paper introduces the Reversible Decoupling Network (RDNet), a novel approach to single-image reflection removal that overcomes key limitations in existing methods. RDNet features a multi-column reversible encoder that preserves hierarchical semantic information by decoupling transmission- and reflection-related features, thus preventing information loss during feature interactions across scales. Additionally, an adaptive transmission-rate-aware prompt generator dynamically adjusts features by learning channel scaling factors from data, which enhances RDNet's generalization and robustness in real-world reflection scenarios.

**Strengths:**

1.This work presents RDNet, which incorporates a multi-column reversible encoder to effectively preserve multi-scale semantic information. By decoupling transmission and reflection features, RDNet minimizes information loss during feature interactions, significantly enhancing reflection removal accuracy.
2.The proposed transmission-rate-aware prompt generator dynamically adjusts feature representations by learning channel scaling factors from the data. This design allows RDNet to achieve strong generalization and robustness across various real-world reflection scenarios.
3.The proposed method outperforms state-of-the-art methods in reflection removal both qualitatively and quantitatively.

**Weaknesses:**

1.The Bidirectional Interaction Level in Figure 2 could benefit from clearer explanation, as the current description in the text is brief and may lead to misunderstandings.
2.The paper compares the proposed method with several existing reflection removal techniques. However, distinct datasets are used in the comparative experiments, which is unnecessary. Additionally, expanding the range of comparison methods would ensure a more comprehensive evaluation, as the current selection may not sufficiently demonstrate the method's effectiveness.

**Questions:**

1.The methods section is not clear enough, for example, line 243 “Bidirectional Interaction Level”, but there is no further explanation and no reference to related work. What is the motivation for using the Bidirectional Interaction Level?
2.It would also be helpful to see results on a unified dataset, including visualizations. Could the authors provide results for the specified methods under consistent training and testing conditions? Such as:
a. Johnson, Justin, Alexandre Alahi, and Li Fei-Fei. "Perceptual losses for real-time style transfer and super-resolution." Computer Vision–ECCV 2016: 14th European Conference, Amsterdam, The Netherlands, October 11-14, 2016, Proceedings, Part II 14. Springer International Publishing, 2016.
b.Wen, Qiang, et al. "Single image reflection removal beyond linearity." Proceedings of the IEEE/CVF Conference on Computer Vision and Pattern Recognition. 2019.
c.Kim, Soomin, Yuchi Huo, and Sung-Eui Yoon. "Single image reflection removal with physically-based training images." Proceedings of the IEEE/CVF conference on computer vision and pattern recognition. 2020.
c.Dong, Zheng, et al. "Location-aware single image reflection removal." Proceedings of the IEEE/CVF international conference on computer vision. 2021.
d.Song, Zhenbo, et al. "Robust single image reflection removal against adversarial attacks." Proceedings of the IEEE/CVF Conference on Computer Vision and Pattern Recognition. 2023.
e.Wang, Mengyi, et al. "Personalized single image reflection removal network through adaptive cascade refinement." Proceedings of the 31st ACM International Conference on Multimedia. 2023.

---

### Official Review · Reviewer_cM2U · 2024-11-02

**Soundness:** 2
**Presentation:** 2
**Contribution:** 2
**Rating:** 3
**Confidence:** 5

**Summary:**

This work focuses on the problem of single-image reflection removal. Its core insight is the proposed Reversible Decoupling Network. The proposed network's core components include a reversible encoder and a transmission-rate-aware prompt generator. The reversible encoder is designed to preserve important information, and the transmission-rate-aware prompt generator dynamically calibrates features to enhance overall effectiveness. The results show that the proposed network performs better than the existing methods.

**Strengths:**

1. Overall, the writing of this manuscript is good and easy to follow.
2.  The ablation study can prove the effectiveness of the proposed methods.
3.  The proposed method outperforms current methods.

**Weaknesses:**

1. The designs of this network are not very new. The proposed network looks more like a complication of existing methods. For example, the multi-scale reversible column encoder, the hierarchy decoder, and the transmission-rate-ware prompt generator have been used in existing methods (e.g., HRNet, ConvNext model).
2. The proposed network looks complex.  There is a lack of comparison in FLOPs and Params in the main comparison (Table 1) and ablation study (Table 3).
3. The paper says " Our method demonstrates superior performance compared to state-of-the-art techniques across a range of real-world benchmark datasets" However, the experimental results of the proposed network were not compared with the latest SOTA methods, such as  "Revisiting Singlelmage Reflection Removal in the Wild (CVPR 2024). "
4. In Figure 3 and Figure 7, there are some artifacts and color distortion in the restored images. For example, the shoes are very different compared to the ground truth.

**Questions:**

1. The PSNR and SSIM primarily focus on pixel-wise evaluation. The authors should also include LPIPS for a more comprehensive perceptual assessment.

2. It is important to provide comparisons regarding FLOPs and parameters. Additionally, running time is a critical factor for real-world applications, so an analysis of the proposed method’s running time should be included.

3. The experimental comparisons should be supplemented with the latest state-of-the-art methods. The authors should also clarify why the results for these comparison methods are comparatively low.

---

### Official Review · Reviewer_DfnT · 2024-11-03

**Soundness:** 3
**Presentation:** 3
**Contribution:** 2
**Rating:** 3
**Confidence:** 5

**Summary:**

This work addresses the problem of single-image reflection removal, introducing a novel Reversible Decoupling Network (RDNet) as its core approach. The RDNet is built around two key components: a reversible encoder, which preserves crucial information, and a transmission-rate-aware prompt generator, which dynamically calibrates features to improve effectiveness. Experimental results demonstrate that the proposed network achieves superior performance compared to existing methods.

**Strengths:**

1. The manuscript is clearly written and easy to follow.
2. The ablation study convincingly demonstrates the effectiveness of the proposed methods.
3. The proposed method achieves improved performance over existing approaches.

**Weaknesses:**

Novelty:

The design of this network is not particularly novel. It appears to be a combination of existing methods. For instance, components such as the multi-scale reversible column encoder, hierarchical decoder, and transmission-rate-aware prompt generator have been utilized in previous works such as HRNet, and ConvNext.


Computational Efficiency:

Given that the network is relatively complex, there’s a lack of discussion around computational efficiency, which is important for real-time or resource-constrained applications. Including information on memory usage, inference speed, and model scalability would be useful for practical implementation.


Artifacts and Color Distortions Problem:

Figures 3 and 7 exhibit artifacts and color distortions in the restored images. For example, some content are changed compared to GT images.

**Questions:**

1. Does the paper analyze failure cases, particularly in challenging situations with complex or severe reflections? Would such an analysis provide a more comprehensive understanding of the model's limitations?
2. Does the study address whether the method can effectively handle high-resolution images, particularly in real-world applications where reflection removal often requires high-resolution processing (e.g., 4K or 8K images)?
3. For other questions,  please refer to the part of the weakness.

---

### Note · Authors · 2024-11-15

I have read and agree with the venue's withdrawal policy on behalf of myself and my co-authors.